# Exploring Helium Ions’ Potential for Post-Mastectomy Left-Sided Breast Cancer Radiotherapy

**DOI:** 10.3390/cancers16020410

**Published:** 2024-01-18

**Authors:** Santa Gabriella Bonaccorsi, Thomas Tessonnier, Line Hoeltgen, Eva Meixner, Semi Harrabi, Juliane Hörner-Rieber, Thomas Haberer, Amir Abdollahi, Jürgen Debus, Andrea Mairani

**Affiliations:** 1Heidelberg Ion-Beam Therapy Center (HIT), Heidelberg University Hospital, 69120 Heidelberg, Germany; 2Division of Molecular and Translational Radiation Oncology, National Center for Tumor Diseases (NCT), Heidelberg University Hospital, 69120 Heidelberg, Germany; 3Department of Radiation Oncology, Heidelberg University Hospital, 69120 Heidelberg, Germany; 4Heidelberg Institute of Radiation Oncology (HIRO), German Cancer Research Center (DKFZ), 69120 Heidelberg, Germany; 5Clinical Cooperation Unit Radiation Oncology, German Cancer Research Center (DKFZ), 69120 Heidelberg, Germany; 6German Cancer Consortium (DKTK), 69120 Heidelberg, Germany; 7Centro Nazionale di Adroterapia Oncologica (CNAO), 27100 Pavia, Italy

**Keywords:** breast cancer, left-sided breast cancer, radiation therapy, particle therapy, proton therapy, helium therapy, VMAT, NTCP model

## Abstract

**Simple Summary:**

This research contributes to the ongoing scientific investigation of the benefits of particle therapy for left-sided breast cancer and presents the potential of the novel application of helium ion therapy. The results obtained reveal a significant improvement in target coverage for both proton and helium ion therapy compared to Volumetric Modulated Arc Therapy (VMAT). Furthermore, particle therapy leads to an increase sparing of surrounding healthy tissues, including the heart, the left anterior descending artery and the ipsilateral lung. Additionally, with the reduction of the low-dose exposure to the right side of the chest, the risk for secondary malignancies with particle therapy is potentially minimized compared to VMAT. Compared to proton therapy, helium could additionally reduce the risk of pneumonitis.

**Abstract:**

Proton therapy presents a promising modality for treating left-sided breast cancer due to its unique dose distribution. Helium ions provide increased conformality thanks to a reduced lateral scattering. Consequently, the potential clinical benefit of both techniques was explored. An explorative treatment planning study involving ten patients, previously treated with VMAT (Volumetric Modulated Arc Therapy) for 50 Gy in 25 fractions for locally advanced, node-positive breast cancer, was carried out using proton pencil beam therapy with a fixed relative biological effectiveness (RBE) of 1.1 and helium therapy with a variable RBE described by the mMKM (modified microdosimetric kinetic model). Results indicated that target coverage was improved with particle therapy for both the clinical target volume and especially the internal mammary lymph nodes compared to VMAT. Median dose value analysis revealed that proton and helium plans provided lower dose on the left anterior descending artery (LAD), heart, lungs and right breast than VMAT. Notably, helium therapy exhibited improved ipsilateral lung sparing over protons. Employing NTCP models as available in the literature, helium therapy showed a lower probability of grade ≤ 2 radiation pneumonitis (22% for photons, 5% for protons and 2% for helium ions), while both proton and helium ions reduce the probability of major coronary events with respect to VMAT.

## 1. Introduction

Breast cancer is the most diagnosed cancer worldwide, with more than 2.3 million new cases in 2020 [1]. Adjuvant radiotherapy is administered following breast-conserving surgery or for locally advanced tumors following mastectomy. However, irradiating the tumor region inevitably involves adjacent healthy tissues, potentially leading to short- or long-term toxicities. For left-sided tumor, organs involvements can be more critical, with clinical implications to the heart.

Recent advances in radiotherapy techniques, such as Intensity-Modulated Radiation Therapy (IMRT) and Volumetric Modulated Arc Therapy (VMAT), have aimed to enhance tumor coverage while sparing nearby organs at risk (OARs) with higher conformality. Nonetheless, they still result in low-dose irradiation throughout the chest area, posing risks of radiation exposure to critical organs such as the heart, contralateral breast and lungs, leading to various toxicities and possibly secondary malignancies [2,3,4,5,6,7,8,9,10].

Since the middle of the last century, charged particle therapy, in particular proton therapy, has been emerging as an alternative option to photon therapy due to its unique depth–dose distribution [11]. This allows us to achieve equal or better target coverage than conventional radiotherapy and simultaneously improves both the uniformity of dose distribution and organs at risk sparing [12]. Other charged particles are used in clinical practice in several worldwide centers such as carbon or helium ions [13]. In 2021, at the Heidelberg ion beam therapy center (HIT), the first patient was treated with raster-scanned helium ions [14]. Helium ions appear to be good candidates for particle therapy due to their intermediate physical and radio-biological properties between proton and carbon ion beams [15]. Compared to proton beams, helium ions show a reduction of lateral scattering, with a lateral spread evolution about 50% smaller [16]. Furthermore, they present a reduced fragmentation tail compared to carbon ions and smaller relative biological effectiveness (RBE) uncertainties [14,15].

Since the early 2000s, proton therapy was suggested as a promising technique for breast cancer treatment and several studies have shown potential benefits for toxicity reductions compared to radiotherapy using photons [10,17,18,19].

So far, the small number of patients involved in proton therapy trials and the consequently few studies on long- or short-term toxicity effects are not sufficient to obtain a complete picture of its clinical advantage over photon therapy [20]. New trials are underway to collect more data, especially in terms of the long-term toxicity landscape [21]. The work presented here contributes to the ongoing proton therapy studies and offers a new perspective about the potential of helium therapy for left-sided breast cancer treatments.

## 2. Materials and Methods

### 2.1. Explorative Treatment Planning with Particle Therapy

A cohort of 10 representative patients (average age of 40 years) treated at the Heidelberg University Hospital for locally advanced node-positive left-sided breast cancer with postoperative conventional radiation therapy, was selected. All of them underwent mastectomy and five of them received immediate breast reconstruction with implants. Treatment was performed with VMAT for a prescribed dose of 50 Gy delivered in 2 Gy/fraction. The thoracic wall and regional lymph node areas, i.e., the axillary lymph node levels I to IV (L1 to L4), the pectoral node (PectoralN) and the internal mammary lymph node (IMN), were irradiated according to current German guidelines [22,23]. Target volume delineation was performed on CT (Computed Tomography) planning images with 3 mm slices thickness in the supine and arms up position and deep-inspiration breath hold technique (DIBH). Target volumes were segmented according to the current European Society for Radiotherapy and Oncology (ESTRO) guidelines [24]. The explorative treatment planning for proton and helium ions was performed with the RayStation treatment planning system (TPS, version 11B, RaySearch Laboratories, Stockholm, Sweden) following the same treatment concept. For protons, as in clinical routine, a RBE of 1.1 was used, whilst for helium ions RBE prediction was performed using the mMKM model [25] with an α/β ratio of 4 Gy [26]. Photon dose is expressed in unit of Gy and charged particle RBE-weighted doses are also expressed in Gy following ICRU recommendations [27,28]; no absorbed dose is reported for ion beam treatment in this work.

Two different gantry angulations were selected for all the patients (35 and 340 degrees) and a 2 cm range shifter was used (airgap between 7 and 15 cm). Due to the limited field size (18 × 18 cm^2^) available at our facility, for each angle two beams were used to cover the target in cranio-caudal direction.

Optimization was performed to reach at least the same clinical goals for OAR as in the VMAT plans while keeping similar or higher target coverage. The following clinical goals were used:at least 95% of prescribed dose to all clinical target volume (CTV);at least 95% of prescribed dose to lymph node regions volume;V_20Gy_ < 8% to the heart;at most 3 Gy as mean dose to the heart;at most 10 Gy as mean dose to the left anterior descending artery (LAD);the LAD V_30Gy_ < 2% and D_2_ < 30 Gy;at most 12 Gy as left lung mean dose;V_20Gy_ < 20% to the left lung;at most 3 Gy as mean doses to contralateral lung and breast.

Dosimetric evaluation was performed among VMAT, protons and helium ion plan. The *evaluated CTV* corresponds to the *initial delineated CTV* (chest wall or implant) without a 3 mm skin region, and is referred to as CTV in this work. CTV and all lymph node region evaluations were performed by extracting D_95_ (where D_x_ is the value of prescribed dose received by *x* percentage of the target volume). Regarding OAR we use the following indices:(a)mean dose value;(b)D_2_, representing the dose that covers 2% of the volume;(c)V_20Gy_, referring to the fractional volume of the organ receiving 20 Gy. In our analysis this index was exclusively considered for the ipsilateral lung.

### 2.2. NTCP Evaluation

To comprehensively address potential acute and long-term toxicity effects of radiation we have employed Normal Tissue Control Probability (NTCP) models, as suggested by Marteinsdottir et al. [29].

#### 2.2.1. Heart

Left-sided breast cancer irradiation entails a greater involvement of the cardiac region as compared to the right side, primarily due to the anatomical positioning of the heart inside the thoracic cavity. This may lead to medical complications which could occur within a certain timeframe after irradiation. 

For heart late toxicity complications, we have considered the following:Cardiac mortality

The probability of incurring cardiac mortality was analyzed with the NTCP seriality model evaluated by the following Equation (1) [30]:(1)$${{\mathrm{NTCP}}_{\mathrm{serial}}=\left\{1-\Pi_{i}^{M}{\left[1-{{P(D}_{i})}^{s}\right]}^{v_{i}}\right\}}^{1/s},$$$D_{i}$ and $v_{i}$ are the dose and volume bins of a differential dose–volume histogram and s is the sensitivity parameter. An s value proximal to unity implies that the organ may be considered as a completely serial structure. Conversely, an s value nearing zero indicates a parallel structure. Each P(Di) is the probability of response of a hypothetic organ i with a volume $v_{i}$, irradiated with a dose $D_{i}\;$ and, as shown in Equation (2), it strictly depends on:(2)PD=2^−exp{eγ(1−DD50)}
where γ is the maximum value of the normalized slope gradient, D_50_ signifies the dose leading in 50% complication risk across the entire heart and D denotes the dose assigned to each individual bin. The clinical endpoint and the parameters were previous studied and calculated by Gagliardi et al. [2]; D_50_ = 52.4 Gy, γ = 1.28 and s = 1.

Heart Valvular Dysfunction (RVD)

RVD NTCP was calculated as in Cella et al. we used the NTCP seriality model (equation 1) applying D_50_ = 32.4 Gy, γ = 0.42, s = 0.99 [3].

Major coronary events

Major coronary event NTCP was predicted using the linear relationship (Equation (3)) by Darby et al. that described a constant increase in risk [4]. The parameter K in Equation (3) was set to 7.4% per heart mean dose in Gray. The Darby NTCP linear model depends on:(3)$$\mathrm{NTC}P_{\mathrm{linear}}=B_{s}\left(1+\mathrm{KX}\right),$$
where X is the mean dose to the heart and the $B_{s}$ parameter that considers the risk of incurring a coronary event in the absence of radiation therapy. Considering our patient cohort characteristic, we fixed $B_{s}$ = 1.9% [4].

#### 2.2.2. LAD

The left anterior descending artery (LAD) encircles cardiac tissue on the left side of the heart. After breast-cancer irradiation therapy it may receive either substantial or a low dose bath dose. For this reason, possibly late toxicity effects and correlation between irradiation and cardiac events [5] or irradiation and coronary stenosis risk [6] have been studied. Moignier et al. [7] proposed to use the odds ratio (OR) logistic model to study the increase in risk of coronary stenosis:Coronary stenosis

The probability of correlation to the mean dose ($D_{\mathrm{med}}$) received was calculated establishing the value of 1.49 for the OR (Equation (4)):(4)$${\mathrm{OR}}_{\mathrm{logistic}}={\mathrm{OR}}^{D_{\mathrm{med}}}.$$

#### 2.2.3. Left Lung

Regarding the pulmonary region, our studies explore the probability of developing radiation pneumonitis (RP), which is mainly caused by the substantial radiation doses received by the lungs during treatment. To assess this complication probability, we opted to follow the methodologies outlined in Martineisdottir et al. [29] employing the $\mathrm{NTC}P_{\mathrm{serial}}$ model (Equation (1)) for the prediction of RP with different grades according to the NCIC-CTC (National Cancer Institute of Canada Common Toxicity Criteria).

RP with grade ≤ 2where the parameters set were D_50_ = 16.3 Gy, γ = 1.08 and s = 0.15 from Rancati et al. [8]; and

RP with grade = 2with D_50_ = 30.1 Gy, γ = 0.966 and s = 0.012 from Gagliardi et al. [9].

### 2.3. Statistical Analysis

Statistical analysis was performed using a two-sided Wilcoxon signed-rank test while comparing the different techniques through the dosimetric parameters, the calculated NTCP or OR. The tests were performed between photon VMAT plans against protons plans, between VMAT and helium ion plans and ultimately between protons and helium ion plans. A *p* value < 0.05 was considered statistically significant.

## 3. Results

Figure 1 displays a comparison between the dose distributions and dose volume histograms (DVH) for photon VMAT, protons and helium ions, for an exemplary patient. Similar to our entire patient cohort, while the CTV coverage is slightly improved with particle therapy plans, a clear dose sparing is achieved for different OARs such as the heart, LAD and left lung.

Table 1 and Appendix A summarize the results obtained for the CTV and IMN coverage using the different treatment modalities. All patients planned with particles exhibited improved coverage compared to VMAT. The CTV D_95_ was achieved in only two patients for VMAT while, for particle therapy, plans D_95_ and D_99_ were reached for all patients. 

For IMN, D_95_ was reached for five patients in VMAT plans, seven in both proton and helium ion plans. However, proton and helium achieved D_90_ in all cases, whereas VMAT achieved it in only seven.

For the axillary and pectoral lymph nodes, the D_95_ was reached for all patients and techniques, except for one VMAT (L2 and L3) and one proton plan (L4). 

### 3.1. Organs at Risk Analysis

Table 2 and Appendix A present a summary of the different dosimetric parameters investigated for the heart, LAD, lungs and contralateral breast.

#### 3.1.1. Heart

The clinical objectives of not exceeding 8% of the volume with more than 20 Gy and 3 Gy as mean doses were fulfilled with all techniques. Mean dose results showed lower values for proton and helium ions compared to VMAT of about 2 Gy less on average. In both cases, they never surpassed 0.6 Gy, revealing a consistent trend of heart sparing in all patients.

The D_2_ was always found to be lower for protons and helium than VMAT, with a difference of approximately 4 Gy and 5 Gy, respectively. Only in one patient was the D_2_ value about 8 Gy for both proton and VMAT scenarios due to the patient’s specific anatomy, whereas helium ions exhibited a smaller D_2_ value (3 Gy).

The NTCP results for the heart are presented in Table 3 and in Figure 2a–c. The risk percentage of cardiac mortality occurrence was close to zero with all techniques and for all patients. The probability of encountering heart valve dysfunction disease was revealed to be lower for particle therapy plans compared to VMAT, with similar NTCP values between protons and helium ions. Lastly, the risk of major coronary events, albeit slightly, was lower for protons and helium plans compared to VMAT.

#### 3.1.2. LAD

The clinical requirement of a LAD mean dose lower than 10 Gy was respected in all plans. Lower mean dose values were found in proton and helium ion plans compared to photon plans. 

Furthermore, the V_30Gy_ < 2% and the D_2_ lower than 30 Gy plan was fulfilled for all VMAT. For nine patients, the D_2_ values were also <15 Gy. Proton and helium plans displayed dosimetric improvements compared to VMAT, with a D_2_ < 9 Gy, except in one patient, who received 17 and 14.5 Gy with proton and helium ions, respectively.

Table 4 and Figure 2 show coronary stenosis OR evaluation for the different treatment modalities, with significant reduction of the OR with particles compared to VMAT.

#### 3.1.3. Left Lung

The mean dose received by the left lung for VMAT treatments was within a 9–12 Gy range (average value about 11 Gy) respecting the defined clinical objectives. For protons, a lower dose bath was found, with mean doses < 9 Gy for each patient and an overall average mean dose value of about 7 Gy. Helium plans displayed an improved sparing of the lung, with mean doses < 7 Gy for each patient and an average mean dose value of 5 Gy. The V_20Gy_ evaluation revealed a lower value for particle therapy plans compared to VMAT plans, mainly for helium ions with a V_20Gy_ < 12.5% for all patients.

In Table 5 and Figure 2, the distribution of NTCP values for radiation pneumonitis occurrence risk is presented. Among ten patients, the results demonstrated that the risk of undergoing a RP with a grade ≤ 2 is much higher for VMAT, with a probability around 20%, with risks of 5% for protons and 2% for helium ions. When considering only grade = 2 RP, VMAT NTCP evaluation showed a mean probability around 2%, higher than the mean complication probability carried out by the proton and helium plan (<0.5%).

#### 3.1.4. Contralateral Lung

One of the clinical objectives included a contralateral lung mean dose of <3 Gy. Among the VMAT treatments investigated, this criterion was met in seven cases, and was below 4 Gy for all patients. For particle therapy plans the mean dose was close to 0 Gy.

#### 3.1.5. Contralateral Breast

The clinical objective of a mean dose of <3 Gy was achieved in seven VMAT plans, with values below 3.5 Gy for all patients. A reduction of the mean dose was observed for charged particle plans, as displayed in Table 2, with a mean dose for the contralateral breast of <0.5 Gy. 

## 4. Discussion

This study explores treatment planning strategies for postoperative radiotherapy in women with locally advanced, node-positive left-sided breast cancer, considering photons (VMAT), protons and helium ions.

While photon techniques (3D-Conformal radiotherapy, IMRT, VMAT) excel in target coverage, trade-offs arise in balancing advantages and disadvantages to protect surrounding organs at risk [20]. Poortmans et al.’s study [31] highlights the benefits of irradiating internal mammary and medial supraclavicular lymph nodes for disease-free and distant disease-free survival. The ten patients presented in this work required regional nodal irradiation, including the IMN. However, the CTV large size together with the inclusion of the regional nodes may lead to a higher dose exposure of surrounding healthy tissues such as LAD, heart and lungs. Therefore, a balance between benefits and potential harms in IMN coverage is crucial in modern radiotherapy [32]. Presently, among the photon techniques, VMAT with deep-inspiration breath hold emerges as the optimal photon technique for post-mastectomy radiotherapy in left-sided breast cancer, ensuring comprehensive target volume coverage with minimal heart and lung exposure [33,34].

The proton plans presented in this work, as well as helium plans, show an improvement of target volume and lymph nodes coverage while minimizing radiation exposure to adjacent organs at risk compared to photon plans, in line with the PTCOG consensus statement on protons for breast cancer treatment suggesting that particle therapy is promising for breast cancer treatment [20,35,36]. These advantageous features are shown in patients with and without breast reconstruction, reinforcing that protons, and in our study helium ions, could be a viable option for different and specific anatomical configurations and indications [20].

According to Darby et al. [4], each Gy of heart mean dose corresponds to an increase of 7.4% in the likelihood of major coronary event induction, with no apparent threshold. The existing literature support approaches aimed at minimizing radiation dosage to every sector of the heart, with the goal of mitigating long-term radiation-induced cardiac morbidity [20]. For all patients, the expected heart mean doses from the proton and helium treatment plans were consistently lower than the mean dose planned with VMAT. We found a decrease in the heart mean dose values for proton and helium therapy compared to VMAT. The significant difference between particle and VMAT plans could translate into a reduction of the occurrence of major coronary events for patients with the same initial cardiac condition. In line with Musielak et al. [35], proton therapy, and helium ions from our study, could be considered a promising approach for mitigating cardiac toxicity risk. The found cardiac mortality NTCP results were close to 0 for all treatments due to the low dose found in the heart, as seen with D_2_ < 10 Gy for all patients and techniques, translating, as described in Gagliardi et al. [2], into a close to zero probability of excess risks. For RVD risk, compared to VMAT, the lower heart dose from particle therapy plans reduced its risk by about 2% according to NTCP seriality model evaluation.

Due to its anatomical position close to the target volumes, the LAD is considered a sensitive critical structure [19], receiving a higher dose compared to the dose received in the heart. This may lead to coronary stenosis risk inductions in accordance with the work of Nilsson et al. [6]. Our results show significant differences in OR between particle therapy and photon radiotherapy, with a reduction of 25%, meaning that the risk of possible toxicity is expected to be lower for particle therapy.

In breast radiotherapy, several factors, such as patient anatomy or tumor localization, can highly impact the extent of ipsilateral lung receiving a radiation dose. The values of all evaluated indices (D_2_, mean dose, V_20Gy_) indicated a downward trend when comparing VMAT with proton therapy and helium therapy. In addition, significant differences are also found between proton and helium ion plans, with improved left lung sparing with helium, with larger differences for the V_20Gy_. As reported in Gokula et al. [37], the mean dose and the V_20Gy_ could be considered as indicators for the risk assessment of radiation pneumonitis. They suggest that the occurrence of RP can be higher if V_20Gy_ > 24%, but with a V_20Gy_ threshold depending on factors such as age or IMN involvement in target definition. According with Wennberg et al. [38], V_20Gy_ < 20% is considered as a threshold to RP occurrences and defined as a clinical goal. In our results (Table 2, Appendix A), the dose received by each patient was higher for VMAT treatment than proton and helium therapy, but still less than 20%. Additionally, we applied NTCP models [8,9] to determine the likelihood of different grade radiation pneumonitis risk induction. We found a significant smaller probability for particle therapy compared to VMAT for RP with grade = 2, but with an even higher difference for RP with grade ≤ 2. In addition, similar to our dosimetric results, helium ion NTCP for RP with grade ≤ 2 appeared to be even lower than the one of protons. 

Unintentional irradiation of the right breast and surrounding tissues during radiotherapy of left breast cancer patients could lead to radiation-induced secondary malignancies. The study of Stovall et al. [39] concludes that women with an age < 40 years and a mean dose to the right breast of more than 1 Gy had a 2.5-fold greater long-term risk of developing second primary contralateral breast cancer. Our results show significantly reduced mean dose values of the contralateral breast compared to VMAT (VMAT 2.89 Gy, proton 0.22 Gy, helium ions 0.04 Gy). Furthermore, similar significant decreases in the mean dose values were observed for the right lung (VMAT 2.56 Gy, proton 0.15 Gy, helium ions 0.11 Gy) that, in line with Paganetti et al. [10], indicate that particle therapy with protons, but also helium ions, represents a good alternative for young patients’ treatments, who could be impacted by long-term toxicity development.

Particle therapy offers a great perspective in treating breast cancer, with a major reduction in risk of coronary stenosis or RP. The smaller low-dose bath in the contralateral lung and breast call for a reduced risk of secondary radiation-induced malignancies, as discussed in Paganetti et al. [10]. These results indicate that particle beams could be of high interest in treating young patients, patients with concurrent lung disease or women with simultaneously applied systemic cardiotoxic therapies [40]. Moreover, helium therapy seems to offer a new opportunity for additional sparing, mainly related to the ipsilateral lung. With the re-appearance of its use for clinical practice and the potential of industrial developments in easing its accessibility [41], helium ions should be considered as a candidate for further research for breast cancer therapy. While this system might not be combined in the early stage together with an isocentric gantry, the increase of interest toward upright position therapy could allow such treatment in a center with only a horizontal room [42].

Our study presents some limitations. It was carried out in silico, using a treatment planning software system, and no patients are yet undertaking particle therapy for left-sided breast cancer in our institute. The current workflow would imply the use of the DIBH technique for CT acquisition and treatments, together with the use of surface guided techniques currently under investigation in our center [43]. Compared to other proton therapy centers, our field size is limited, which leads to the need for additional beams and isocenters that would increase the treatment time for the patients. Furthermore, while our facility was mainly designed for heavy ions, the proton beam focus width for low energy is known to be larger than cyclotron-based centers. However, despite this fact, and most probably linked to the use of a smaller range shifter (only 2 cm) compared to other facilities for shallow target treatments (between 5 and 7 cm), no compromise in the quality of our plan between target coverage and sparing of OAR was needed, and our dosimetric and NTCP predictions were similar to other published work [44,45,46,47,48], as shown in Table 6 and Table 7. 

Dynamic collimation system (DCS) with proton spot scanning might reduce the difference in ipsilateral lung dose found between proton and helium ions by reducing the proton lateral dose penumbra [49]. Current state-of-the art proton DCS, such as the Mevion proton therapy system with Hyperscan, could improve the lateral dose fall-off for proton breast treatment for the investigated therapeutic low-energy range. However, with the current system, this feature comes at the cost of a larger distal dose fall-off compared to classic proton beam systems, which might reduce the expected advantages on the ipsilateral lung dose, as well as a reduced homogeneity in the target [49,50]. Nevertheless, further development in collimation strategies would be beneficial for both the proton and helium ion therapies.

The results of this study for protons were estimated for a fixed RBE of 1.1. According to several studies [29,51,52,53] and reported in the PTCOG consensus statement on proton breast cancer treatment [20], a variable RBE for protons should be considered because the results could suffer of an underestimation of the RBE-weighted dose delivered at the distal edge of the target. Consequently, this could lead to an underestimation of the predicted risk occurrences. This could impact the NTCP results, as shown in Marteinsdottir et al. [29], where, using a variable RBE model, a potential increase in RP (grade ≤ 2) and OR for coronary stenosis was found, bringing their NTCP results close to the one of photons. Other endpoints were investigated in the literature, where proton therapy can lead to short- or long-term toxicity effects, such as possible rib fractures and skin toxicity. The in silico study by Tommasino et al. [54] showed a comparable dose-to-skin value between photon and proton therapies, but the work of DeCesaris et al. [55] highlighted higher acute grade ≥ 2 radiation dermatitis compared to photons. Fattahi et al. [56] evaluated a short-term follow-up of nineteen patients and concluded that protons could possibly increase the rate of rib fractures. This higher rate of rib fractures could be linked to the increase in LET and RBE at the distal end of the beam [57].

However, all the plans with helium ion beams were carried out using the mMKM model for predicting the RBE-weighted dose distribution, meaning that the dosimetric and NTCP predictions will keep their favorable results in comparison to VMAT and are expected to be improved compared to protons using a variable RBE model. RBE models, such as mMKM, are known to be sensitive to their initial model parameters. However, a sensitivity study [58] suggested a relatively small impact (about 5%) of these parameters on the RBE predictions for a known α/β. An α/β of 4 Gy was used in this work, which might impact the results for the heart and lung. While there is no clear value in the literature, the range of α/β is expected to be close to 3–4 for the lung and heart [59,60,61,62]. Changing this value to 3 would lead to a potential increase of about 6%, which would not change the benefits of helium ion therapy over photons in general, or protons regarding lung sparing. Similarly, with RBE variation of the order of 5–6% for the target volume (stemming from the α/β value or β value), such changes are not expected to bring change in tumor control probability.

Initial evaluations of the robustness of the treatment plans against patient position and range uncertainties were performed for two representative patients (with and without breast reconstruction). The evaluation against a 3% range uncertainty did not impact the CTV coverage or OAR goals for either photons or particle therapy plans. The robust evaluation for a 5 mm patient position uncertainty impacted the CTV coverage in the particle therapy plans, mainly for the patient with breast reconstruction. However, the particle therapy CTV coverages remained, in all scenarios and patients, higher than the one from VMAT plans. Due to the specific need for four beams in our facility, adding robustness optimization goals could help in reducing the individual beam dose gradient and improving the robustness against patient position uncertainty, without impact on the surrounding OAR. Further studies are warranted to investigate in depth robust optimization planning and evaluation for protons and helium ions. 

There is a wide range of available fractionation schemes in the postoperative irradiation of women with breast cancer. Our study represents the results of a normofractionated radiotherapy scheme, which was chosen for a better comparison with other center expertise and as national guidelines still prefer normofractionation for patients who need regional nodal irradiation due to a lack of long-term data and lack of prospective randomized controlled trials. However, the international consensus advocates a trend for hypofractionated regimes, for which the dosimetric effects of particle therapy still have to be evaluated [63,64]. However, the general advantages of particle therapy over photons are still expected to hold true. Among the patient cohort, a net advantage was found in CTV coverage when comparing the particle therapy plan against a photon plan for thorax wall treatment. These results also suggest that particle therapy using a proton and helium beam could be a good candidate in treating men with breast cancer [65]. 

While particle therapy sounds appealing for post-mastectomy left-side breast cancer, the costs and scarcity of the centers are reducing their potential impact. Nevertheless, the PTCOG Breast Cancer subcommittee recommends consideration for enrolment in clinical trials and that studies have shown that proton therapy could be cost effective for women with left breast cancer with regional nodal involvement [20,66]. The number of centers that could provide helium ions in clinical practice is rather limited, confined to synchrotron-based facilities. Consequently, among the indications for helium ion therapy, breast cancer treatment might not be among the first indications to be handled in comparison to others where the sharp helium ion lateral dose penumbra could help in reducing treatment related sequelae, such as in head, head and neck or pediatric tumors. However, with the upcoming novel superconducting cyclotron or synchrotron [67,68], reducing the facility footprint and cost, helium ions could be made more available for clinical practice. For future ion facilities without access to protons [67], the results of this study could provide initial insight into the possibilities of left-sided breast cancer treatments with helium ions.

Despite the limitations of this work, we believe that this in silico study may offer, not only support for ongoing research on proton therapy benefits for breast cancer treatment, but also a new valuable input for an alternative direction in the use of particles in radiotherapy for cancer treatment with helium ion beams.

## 5. Conclusions

This investigation offers an initial perspective of the advantages of helium therapy for the treatment of postoperative radiotherapy for locally advanced left-sided breast cancer and sustains the ongoing research for proton therapy. Each patient planned with charged particles showed an improvement in target and especially internal mammary chain coverage as well as OAR sparing compared to VMAT plans.

Particle therapy with protons and helium ions could offer a great advantage over photons for young women due to the decrease in the probability of pulmonary and cardiac long-term toxicities, as well as contralateral breast cancer induction. We believe that helium therapy needs to be explored further to broaden our understanding of its potential benefits. 

## Figures and Tables

**Figure 1 cancers-16-00410-f001:**
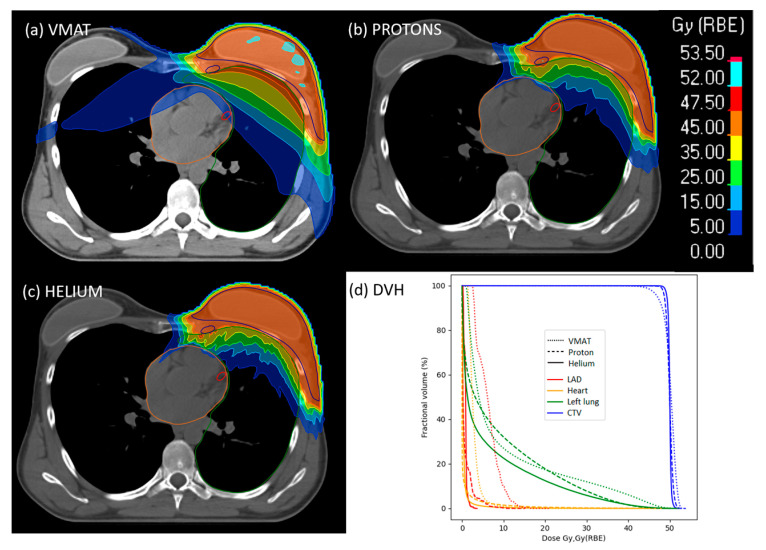
Example of dose distributions for left-sided breast cancer obtained with (**a**) VMAT, (**b**) proton, (**c**) helium ions. (**d**) Presents the dose volume histograms (DVH) for a single patient with the clinical target CTV in blue, LAD in red, heart in orange, left lung in green. The data from the VMAT plan are represented with dotted lines, from the proton plan with dashed lines and for the helium with solid lines. All the other DVHs can be found in the Appendix A.

**Figure 2 cancers-16-00410-f002:**
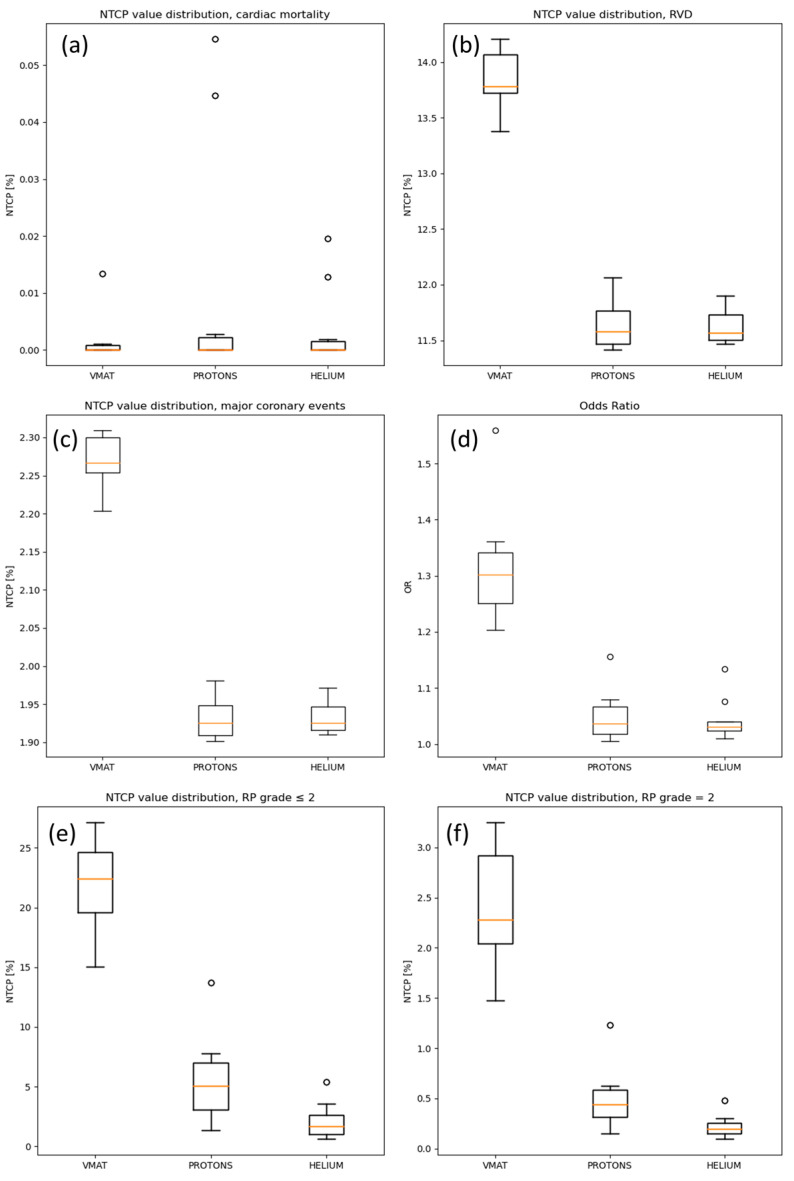
Boxplot distribution of the NTCP predictions for the different investigated clinical endpoints in the patient cohort: (**a**) heart cardiac mortality, (**b**) RVD, (**c**) heart major coronary events, (**d**) LAD coronary stenosis risk occurrence, (**e**) left lung RP grade ≤ 2 and (**f**) RP grade = 2. NTCP: normal tissue control probability, RVD: heart valvular dysfunction, LAD: left anterior descending artery, RP: radiation pneumonitis.

**Table 1 cancers-16-00410-t001:** Clinical target volume (CTV), internal mammary lymph node (IMN), axillary lymph node levels I to IV (L1 to L4) and pectoral node (PectoralN) chain coverage percentage for VMAT (Volumetric Modulated Arc Therapy), proton and helium ion plans. * Is displayed when either proton or helium ion plans present a significative difference against the photon plans (*p* < 0.05). *p*-Value results can be found in Table A1.

Coverage %							
Technique	CTV	IMN	L1	L2	L3	L4	PectoralN
VMAT	93.1 (90.6–95.4)	88.6 (68.2–99.7)	98.7 (97.5–100.0)	97.8 (93.6–100.0)	98.9 (93.5–100.0)	99.1 (97.8–100.0)	99.2 (96.0–100.0)
Proton	99.9 * (99.9–100.0)	94.7 (90.0–97.4)	100.0 (100.0–100.0)	100.0 (100.0–100.0)	100.0 (100.0–100.0)	97.8 (92.8–99.6)	100.0 (100.0–100.0)
Helium	99.8 * (99.8–100.0)	95.7 (89.7–100.0)	100.0 (100.0–100.0)	100.0 (100.0–100.0)	99.5 (96.6–100.0)	99.0 (98.2–100.0)	100.0 (100.0–100.0)

**Table 2 cancers-16-00410-t002:** Dosimetric analysis for the different investigated techniques (Volumetric Modulated Arc Therapy (VMAT), proton and helium ions). Mean values (and ranges) are presented for: the mean dose and the D_2_ values for heart, left anterior descending artery (LAD), lungs and contralateral breast; V_20Gy_ for the left lung.

	Dosimetric Analysis		
	VMAT	Proton	Helium
Heart			
mean dose [Gy]	2.61 (2.16–2.91)	0.22 * (0.01–0.57)	0.23 * (0.07–0.51)
D_2_ [Gy]	6.75 (4.53–9.25)	2.71 * (0.05–7.79)	1.46 * (0.34–3.38)
LAD			
mean dose [Gy]	5.64 (3.87–9.28)	0.96 * (0.10–3.02)	0.86 * (0.20–2.63)
D_2_ [Gy]	11.88 (5.46–27.19)	5.54 * (0.73–17.09)	4.05 *^‡^ (0.36–14.52)
Left Lung			
mean dose [Gy]	10.68 (9.35–11.75)	6.37 * (4.39–8.86)	4.93 *^‡^ (3.60–6.63)
D_2_ [Gy]	44.97 (41.85–47.16)	38.25 * (34.03–40.73)	36.00 *^‡^ (30.94–40.51)
V_20Gy_ (%)	18.04 (15.57–19.21)	12.00 * (7.98–17.79)	8.59 *^‡^ (6.04–12.47)
Right Lung			
mean dose [Gy]	2.56 (1.20–3.52)	0.15* (0.01–0.35)	0.11 * (0.02–0.30)
Contralateral Breast			
mean dose [Gy]	2.89 (2.52–3.36)	0.22 * (0.07–0.42)	0.04 *^‡^ (0.00–0.08)

*** Is displayed when either proton or helium ion plans present a significative difference against photon plans (*p* < 0.05). ‡ Is displayed when proton and helium ions present significative differences between their plans. *p*-Value results can be found in Table A1.

**Table 3 cancers-16-00410-t003:** NTCP mean values (and range) for assessing the probability of the occurrence of late toxicity effects due to heart dose received as major coronary events, cardiac mortality or RVD. NTCP: Normal tissue complication probability; VMAT: Volumetric Modulated Arc Therapy; RVD: heart valvular dysfunction.

	NTCP (%)		
Clinical Endpoints	VMAT	Proton	Helium
Cardiac mortality	0.0 (0.0–0.0)	0.0 (0.0–0.1)	0.0 (0.0–0.0)
RVD	13.8 (13.4–14.2)	11.6 *** (11.4–12.1)	11.6 *** (11.5–11.9)
Major coronary events	2.27 (2.20–2.31)	1.93 *** (1.90–1.98)	1.93 *** (1.91–1.97)

*** Is displayed when either proton or helium ion plans present a significative difference against photon plans (*p* < 0.05). *p*-Value results can be found in Table A2.

**Table 4 cancers-16-00410-t004:** Odds ratio mean values and range for the coronary stenosis endpoint for the three treatment modalities. VMAT: Volumetric Modulated Arc Therapy.

	Odds Ratio		
Clinical Endpoint	VMAT	Proton	Helium
Coronary stenosis	1.31 (1.20–1.56)	1.05 *** (1.00–1.15)	1.04 *** (1.01–1.13)

*** Is displayed when either proton or helium ion plans present a significative difference against photon plans (*p* < 0.05). *p*-Value results can be found in Table A2.

**Table 5 cancers-16-00410-t005:** NTCP evaluation results for radiation pneumonitis. NTCP: Normal tissue complication probability; VMAT: Volumetric Modulated Arc Therapy.

	NTCP (%)		
Clinical Endpoints	VMAT	Proton	Helium
Left Lung			
RP grade ≤ 2	21.9 (15.0–27.1)	5.4 * (1.4–13.7)	2.1 *^‡^ (0.6–5.4)
RP grade = 2	2.4 (1.5–3.3)	0.5 * (0.2–1.2)	0.2 *^‡^ (0.1–0.5)

* Is displayed when either proton or helium ion plans present a significative difference against photon plans (*p* < 0.05). ‡ Is displayed when proton and helium ions present significative differences between their plans. *p*-Value results can be found in Table A2.

**Table 6 cancers-16-00410-t006:** Dosimetric indices comparison among our work and some of the most recent results published on proton therapy for breast-cancer treatments [29,44,45,46,47,48]. VMAT: Volumetric Modulated Arc Therapy, IMPT: Intensity Modulated Proton Therapy, LAD: left anterior descending artery.

This Work				
	Heart	LAD	Left Lung	
	Mean Dose [Gy]			V_20Gy_ [%]
VMAT	2.61 (2.16–2.91)	5.64 (3.87–9.28)	10.68 (9.35–11.75)	18.04 (15.57–19.21)
Proton	0.22 (0.01–0.57)	0.96 (0.10–3.02)	6.37 (4.39–8.86)	12.00 (7.98–17.79)
Helium	0.23 (0.07–0.51)	0.86 (0.20–2.63)	4.93 (3.60–6.63)	8.59 (6.04–12.47)
**Gao et al.** (2023)—Mayo Clinic [44]				
IMPT	0.62 (0.37–0.89)	2.11 (1.15–3.43)	7.54 (7.01–8.43)	14 (12.0–15.0)
**Loap et al.** (2021)—Institute Curie [45]				
VMAT	3.3	6.4	11.0	
Proton	0.4	0.4	7.9	
**Cartechini et al.** (2020)—TIFPA [46]				
VMAT			16.4 (14.8–18.0)	
Proton			8.4 (8.1–8.7)	
**De Rose et al.** (2020)—Humanitas Research Hospital and Cancer Center [47]				
VMAT	3.9 ± 0.9		10.8 ± 1.1	
IMPT	0.4 ± 0.3		6.2 ± 0.8	
**Jimenez et al.** (2019) Massachusetts General Hospital, Boston [48]				
Proton	0.5 (0.10–1.70)	1.16 (0.09–12.00)	7.72 (2.39–13.80)	14.5 (8.76–22.24)
**Marteinsdottir et al.** 2021 [29]				
VMAT	3.9 (3.3–5.5)	6.2 (5.5–12.8)	12.8 (10.1–14.1)	22.9 (15.4–26.2)
Proton	0.5 (0.2–1.1)	0.5 (0.1–2.6)	8.7 (5.8–9.1)	16.3 (9.3–18.2)

**Table 7 cancers-16-00410-t007:** NTCP value comparison among the results from the Marteinsdottir et al. [29] recent publication on proton therapy for breast-cancer treatments. VMAT: Volumetric Modulated Arc Therapy, LAD: left anterior descending artery, OR: Odds Ratio.

This Work			
	NTCP Values		
Clinical Endpoints	VMAT	Proton	Helium
cardiac mortality [%]	0.0 (0.0–0.0)	0.0 (0.0–0.1)	0.0 (0.0–0.0)
RVD [%]	13.8 (13.4–14.2)	11.6 (11.4–12.1)	11.6 (11.5–11.9)
major coronary events [%]	2.27 (2.20–2.31)	1.93 (1.90–1.98)	1.93 (1.91–1.97)
coronary stenosis (OR)	1.31 (1.20–1.56)	1.05 (1.00–1.15)	1.04 (1.01–1.13)
RP grade ≤ 2 [%]	21.9 (15.0–27.1)	5.4 (1.4–13.7)	2.1 (0.6–5.4)
RP grade = 2 [%]	2.4 (1.5–3.3)	0.5 (0.2–1.2)	0.2 (0.1–0.5)
**Marteinsdottir et al.** 2021 [29]			
cardiac mortality [%]	0.0 (0.0–0.1)	0.0 (0.0–0.0)	
major coronary events [%]	2.1 (2.0–2.2)	1.7 (1.6–1.8)	
coronary stenosis (OR)	1.5 (1.3–2.7)	1.1 (1.0–1.2)	
RP grade ≤ 2 [%]	32.6 (16.7–40.9)	12.1 (3.0–14.1)	
RP grade = 2 [%]	4.2 (1.9–5.9)	1.1 (0.3–1.3)	

## Data Availability

The data presented in this study are available on request from the corresponding author.

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
