# Peer review of "Exploring Helium Ions’ Potential for Post-Mastectomy Left-Sided Breast Cancer Radiotherapy"

_cancers, 2024, doi:10.3390/cancers16020410_

Round 1

Reviewer 1 Report

Comments and Suggestions for Authors

General comments:

The manuscript is a typical treatment planning study, comparing advanced photon beam therapy with proton and helium ion therapy, for left-sided breast cancer patients. The study was based on only 10 patients, which I consider as lower minimum number for treatment planning studies in general. 

Given the large number of published treatment planning studies comparing photons and protons, the results come at no surprise. Helium ion therapy for breast cancer is certainly novel, but again based on the publications on Helium ions the results are not surprising. 

The planning study deals with a specific breast cancer patient cohort, i.e. RT in a post-mastectomy setting. This should be mentioned in the title as well. There are other planning studies on left sided breast cancer patients with protons, but in a different setting (e.g. partial breast irradiation and other techniques for early stage tumors).

The manuscript is generally well written and clear; when looking at the well conducted NTCP modelling, the applied methods go even beyond other planning comparisons.

In my point of view, the authors missed two important clinical / practical aspects, that should at least be discussed. First, there is a trend towards hypofractionation for breast cancer patients, even for post-mastectomy RT. The assumed fractionation scheme get outdated. Next, aspects of deep inspiration breath hold for particle therapy +/- IGRT or surface guidance need to be addressed, otherwise this planning study remains a purely academic exercise. I recommend to include to consider the following publications in the context of the comments made above:

-        Mutter RW, et al Proton Therapy for Breast Cancer: A Consensus Statement From the Particle Therapy Cooperative Group Breast Cancer Subcommittee. Int J Radiat Oncol Biol Phys. 2021;111(2):337-359. doi: 10.1016/j.ijrobp.2021.05.110. 

-        Alterio D, et al Hypo-fractionated proton therapy in breast cancer: where are we? A critical review of the literature. Breast Cancer Res Treat. 2022;192(2):249-263. doi: 10.1007/s10549-022-06516-4.

Given the current results with photon therapy for breast cancer patients and the costs related to particle therapy, I question the (clinical) impact of Helium ions for breast cancer in general. Why not exploring the added value of Helium ions for more established indications of particle therapy?

Specific comments:

Line 315: wording “larger sparing” should be replaced by “improved sparing”

Line 343: should be “study” not “studies”

Lines 476-478: first sentence of section conclusion reads like a summary and not like a conclusion. Can be skipped in my view.

Reviewer 2 Report

Comments and Suggestions for Authors

Cancers invited me to review the manuscript "Exploring Helium Ions Potential for Left-Sided Breast Cancer Radiotherapy", which describes an in-silico planning study focusing on the possible reduction of side effects. The manuscript is quite comprehensive and generally well written. However, a number of major modifications should be considered before considering this study for publication:

1)      As pointed out by the authors in lines 446-449, the uncertainty of RBE should be considered. However, the authors discussed the ramifications of RBE uncertainties only for NTCP. What could be the consequences for TCP?

2)      Obviously, plan robustness was neglected. Although the consistent plan optimization for all 3 modalities has some methodological justification, I would expect a clearly larger impact of set-up uncertainties for both particle modalities. Robust optimization would have washed out the lateral dose gradient of particle fields thereby reducing their physical advantage. Range uncertainties have obviously been ignored. These aspects introduce a bias in favor of particle fields (in addition to item 1).

3)      L. 40, 81: I guess that lateral scattering is the key advantage of He beams and the reduced range straggling is clinically negligible (the latter should be compared to the range uncertainties, see 2). I' surprised to find straggling in prominent position in the abstract. Please provide some numbers to quantify the gain with He regarding these quantities.

4)      L. 417-426: The authors are too optimistic regarding the clinical availability of He beams. I think that the IBA project (Ref. 25) has been on hold for a few years. The current study also employs an ion gantry, which further increases the necessary investment. Given the very, very limited available treatment slots, should the indication spectrum be detailed/narrowed further?

5)      Regarding items 3 and 5 and lines 432-437: there are many projects dealing with apertures and proton PBS, which also sharpen the lateral penumbra. Dynamic collimators could maybe compete with He beams, but are either at the prototype stage or a combined with large spots (MeVion). Although static apertures can only trim the dose at the outer edges (and are not ideal for complex target volumes), they are quite effective to reduce the dose a few cm away from the field edge (potentially the heart in this application?). So there is a potential for He beams, but there is a possible technical alternative.

6)      Interestingly, the PTCOG consensus guideline about breast irradiation (Mutter et al., Int J Radiat Oncol Biol Phys. 2021 October 01; 111(2): 337–359. doi:10.1016/j.ijrobp.2021.05.11) is not cited. There is an overlap with some descriptions and discussions of the current work. The authors should check their coherence with the guideline and check if the text can be shortened by referencing to this guideline.

Moderate issues:

-          L. 112: The description of RBE-weighted dose for protons is not according to ICRU report 78, which recommends either "RBE" in the index, e.g. "DRBE,prescr = .." or together with the unit "Gy(RBE)". The description deviates between the main text and some figures (e.g. Gy(RBE) in the label of Fig. 1d).

-          Eq. 4: D_med in the exponent should have no unit (but is Gy …)

Minor issues:

-          The description is not always self-consistent, e.g. "PROTON" in Tab. 2 and "Protons" in Tab. 1.

-          Caption of Tab. xplains "OR" which is not used.

-          Axis labels and tics are hard to read in the print-out, e.g. Fig. 2.

-          L. 109: "For protons …"

-          L 160: "as shown"

-          L 239: "green. The"

-          L. 267: "for the heart"

-          L. 315: "Helium plans"

-          L. 376: "were close"

-          L. 402: "similar to"

-          L. 420: "particle beams"

Round 2

Reviewer 1 Report

Comments and Suggestions for Authors

Thank you for the thorough revision. All my previous comments have been addressed in an adequate manner.

Author Response

We would like to thanks a lot the reviewer for his comments which helped us to improve the manuscript.

Reviewer 2 Report

Comments and Suggestions for Authors

This report concerns the review the revised manuscript "Exploring Helium Ions Potential for Post-Mastectomy Left-Sided Breast Cancer Radiotherapy". I identified the following issues:

·       -The rebuttal of the authors regarding the collimators in combination with proton fields is not convincing. In my remark "few cm away from the field edge" of the first report I referred to static apertures. Dynamic apertures might be able to sharpen the lateral penumbra at the field edge. Thus, they would provide dosimetric benefits for the ipsilateral lung.
Furthermore, the authors contradict themselves: In their rebuttal the heart is ranked as secondary goal (compared to lung), while they still state in the conclusions ""the decrease in probability of contralateral breast cancer induction as well as cardiac and pulmonary long-term toxicities.", i.e. cardiac and pulmonary toxicities are on the same level.

·        - The rebuttal to me regarding the robust optimization is acceptable. However, some readers may ask similar questions. Thus, a summary of the evaluation of the robustness aspects should be included in the manuscript.

·        - Both reviewers asked the authors to incorporate "Mutter et al., Int J Radiat Oncol Biol Phys. 2021 October 01; 111(2): 337–359. doi:10.1016/j.ijrobp.2021.05.11)". The inclusion of this study is reduced to a single citation instead of a full incorporation into the discussion.

·        - The reviewers suggested to detail/narrow "the indication spectrum" and questioned "the (clinical) impact of Helium ions for breast cancer in general." Therefore, I expected a more critical discussion in the frame of the revision. This was not done.

Author Response

We would like to thanks a lot the reviewers for giving us further insights to improve the manuscript. We attached our point-by-point response. 
